# Cardiovascular Precision Medicine and Remote Intervention Trial Rationale and Design

**DOI:** 10.3390/jcm13206274

**Published:** 2024-10-21

**Authors:** Deborah Reynolds, Rachel A. Annunziato, Jasleen Sidhu, Gad Cotter, Beth A. Davison, Koji Takagi, Sarah Duncan-Park, David Rubinstein, Eyal Shemesh

**Affiliations:** 1NYC Health + Hospitals/Elmhurst, Queens, New York, NY 11373, USA; 2Icahn School of Medicine at Mount Sinai, 1 Gustave L Levy Place, New York, NY 10029, USA; 3Department of Psychology, Fordham University, Bronx, New York, NY 10458, USA; 4Tulane University School of Medicine, New Orleans, LA 70112, USA; 5Momentum Research, 1426 East NC Highway 54, Suite B, Durham, NC 27713, USA; gadcotter@momentum-research.com (G.C.); bethdavison@momentum-research.com (B.A.D.);; 6Mindich Child Health and Development Institute, Icahn School of Medicine at Mount Sinai, New York, NY 10029, USA

**Keywords:** precision medicine, cardiovascular illness, behavioral health intervention, adherence

## Abstract

**Background:** It has recently been shown that excessive fluctuation in blood pressure readings for an individual over time is closely associated with poor outcomes, including increased risk of cardiovascular mortality, coronary heart disease and stroke. Fluctuations may be associated with inconsistent adherence to medical recommendations. This new marker of risk has not yet been incorporated into a monitoring and intervention strategy that seeks to reduce cardiovascular risk by identifying patients through an algorithm tied to their electronic health record (EHR). **Methods:** We describe the methods used in an innovative “proof of concept” trial using CP&R (Cardiovascular Precision Medicine and Remote Intervention). A blood pressure variability index is calculated for clinic patients via an EHR review. Consenting patients with excessive variability are offered a remote intervention aimed at improving adherence to medical recommendations. The outcomes include the ability to identify and engage the identified patients and the effects of the intervention on blood pressure variability using a pre–post comparison design without parallel controls. **Conclusions:** Our innovative approach uses a recently identified marker based on reviewing and manipulating EHR data tied to a remote intervention. This design reduces patient burden and supports equitable and targeted resource allocation, utilizing an objective criterion for behavioral risk. This study is registered under ClinicalTrials.gov Identifier: NCT05814562.

## 1. Introduction

Globally, coronary heart disease (CHD) is recognized as the leading cause of death and is predicted to remain so for the next ten years. Worldwide in 2019, ~18 million deaths were attributed to CHD as well as 38% of premature deaths, defined as death under the age of 70, mostly due to heart attacks or stroke [1]. The control of hypertension and hypercholesterolemia is key to reducing the risk of adverse cardiovascular outcomes in patients with CHD, and behavioral and demographic constructs, including nonadherence to medical recommendations, are the most important reasons for adverse outcomes in patients with hypertension [2,3].

Recently, increasing attention has been given to the role of blood pressure (BP) variability as an independent prognostic indicator in various studies [4,5,6,7,8]. This variability has also been linked to a range of health consequences, including cardiovascular disorders, chronic renal disease, cognitive deterioration, and psychiatric conditions [9]. A recent study observed that 33.2% of hypertensive patients exhibited substantial BP variability [10]. Various factors contribute to fluctuations in BP, such as improper dosage or titration of antihypertensive therapy, increased arterial stiffness, age, decreased adherence to therapy, and BP measurement errors [11]. Therefore, controlling BP variability is likely to prove important not only for hypertension management but also for preventing adverse cardiovascular events [4,5,6,7,8]. The increasing use of wearable technology that can monitor blood pressure more frequently [12] may make it easier to recognize significant fluctuations, but it is unclear what approach, if any, can be taken to reduce such variability.

More than two decades ago, our group noted that excessive fluctuation in certain biological indices (such as medication blood levels) reflects a behavioral risk (nonadherence to a prescribed treatment) and predicts poor outcomes [13,14]. The idea that fluctuation reflects behavioral risk is now considered an established fact in transplant medicine [15,16]. More recently, other medical fields have become interested in evaluating fluctuations in biological indices such as cholesterol levels [17,18] and BP readings [19,20,21]. Fluctuations in those parameters are also closely associated with poor outcomes. It is likely that such fluctuations, especially when excessive, are also related to nonadherence to medications that are supposed to mitigate the identified risk (e.g., antihypertensives) [19,20,21,22]. The way fluctuation is measured by our established index in transplant medicine and cardiovascular studies is similar: as a calculation of the standard deviation (SD) of a set of readings for the same patient over time. A higher SD, or higher coefficient of variation (CoV, the SD divided by the mean, presented as a percentage), denotes more variability between individual readings.

Despite ample evidence that excessive fluctuation in BP readings for an individual over time is closely associated with poor outcomes [23,24,25,26,27,28,29], we are unaware of prospective, dedicated efforts to use this relatively new insight in order to design novel approaches to interventions aiming to reduce such fluctuation. The CP&R trial (Cardiovascular Precision Medicine and Remote Intervention) is, to our knowledge, the first trial seeking to use this BP fluctuation metric as a marker to identify “at-risk” patients via surveillance of their electronic health record (EHR).

The CP&R protocol employs a minimally burdensome approach in an urban center with one of the world’s most diverse patient populations [30,31,32,33]. It targets a select group of at-risk yet currently stable patients, identified through an EHR review. The intervention utilizes a remote “core” of interventionists who address avoidance coping, an issue previously studied in transplant recipients as well as patients with cardiovascular illnesses [34]. The aims include assessing both the implementation feasibility as well as the potential for having an impact on the fluctuation metric during the intervention.

## 2. Methods

### 2.1. Study Design

CP&R is a single-center, open-label, single-group assignment study, registered under ClinicalTrials.gov Identifier: NCT05814562. It aims to enroll patients with hypertension and hyperlipidemia, who are identified through an EHR algorithm. The coefficient of variation (CoV) of at least three systolic BP readings over the preceding year was calculated, and patients with CoVs > 10% (indicating higher-than-average fluctuation [35]) were enrolled upon providing informed consent. This study has received ethical approval from Mount Sinai’s IRB and Elmhurst’s research committee. Enrolled patients were provided with weekly remote sessions for the first month and biweekly sessions for the second and third months. The patients’ BP was monitored via a home-based blood pressure monitoring device at each session. BP fluctuations (average changes and changes in the CoV) as well as changes in total cholesterol levels were evaluated at the end of the intervention. For the purpose of the primary analysis of systolic blood pressure fluctuation, the termination of the intervention was at 12 weeks post-enrollment with a “window” of 2 months. Other analyses might use other timeframes, for example, the actual date of the last interaction or the date of the last recorded blood level value.

### 2.2. Study Population and Eligibility Criteria

The Cardiovascular Clinic at Elmhurst Hospital serves one of the most diverse neighborhoods in the world. This allows for the recruitment of a racially, ethnically, and economically diverse pool of subjects who are likely to—on the one hand—pose particular challenges to recruitment and engagement (due to potential language or cultural barriers), but, on the other hand, are perhaps most likely to benefit from the intervention [36]. The availability of language interpretation services and multilingual staff makes it possible to cater to this diverse population. Patients are seen irrespective of their immigration and financial status, enabling some of the poorest patients to obtain the care they need. The hospital pharmacy also provides affordable medications to patients without insurance, ensuring that cost is not a barrier to standard care [37].

The inclusion and exclusion criteria for the study are presented in Table 1. Patients had to be over 18 years of age and under care for hypertension and hypercholesterolemia at the cardiology clinic and have been prescribed medications for these conditions for at least 6 months. Eligibility also required that the patients were reachable via phone or the internet and spoke English or Spanish. Exclusion criteria included psychiatric disorders, complicated additional medical conditions, recent thromboembolic events, or hospitalization in the period of 6 months pre-enrollment. Recently hospitalized patients and patients with mental health disorders or other complex medical comorbidities were excluded in order to ensure that patients enrolled in the trial were relatively stable and would not have been necessarily “flagged” for increased surveillance in the absence of the variability marker criterion. Informed consent was obtained, and participants could withdraw without affecting their ongoing medical care.

### 2.3. Intervention

Our group developed a remote intervention paradigm in which a trained interventionist contacts (via telephone or internet chat) patients on a set schedule to review barriers to medication-taking and remedies for those barriers. The tailored session typically lasts 10–30 min and follows a manual. Such an intervention has been successful in several small studies in different transplant settings, where we showed that it reduces the variability in the chosen index. Our previous work at Elmhurst Hospital Center has supported the safety and efficacy of a general approach to addressing avoidance and posttraumatic stress symptoms as important barriers to adherence [42], and this work informed some of the content of the intervention; posttraumatic stress with avoidance symptoms has been shown to be prevalent in the same clinic [36].

### 2.4. Study Outcomes

This pilot proof-of-concept study focuses on evaluating both the feasibility and the potential impact of a remote intervention aimed at patients with high BP variability in order to provide information for the design of a confirmatory investigation. Implementation data are crucial because of the focus on a subset of a population who are likely to be nonadherent to medical recommendations and therefore presumably are also harder to engage (harder to approach because they are more likely to miss clinic visits, harder to obtain consent from, and harder to successfully engage over a period of time). The study’s implementation aims are to (1) define which proportion of ‘at-risk’ patients exist in a given clinic via EHR reviews performed at baseline; (2) recruit a subset of these identified ‘at-risk’ patients who agree to participate and define what percentage of those approached agree to participate; and (3) define what proportion of those who are recruited can be engaged in at least three complete remote intervention and monitoring sessions over a 12-week period. To assess the potential impact of the intervention (for the purpose of power assumptions for a confirmatory study, if justified based on the data in this pilot), the study monitored changes in the BP variability index (BPSDV, the SD of consecutive readings), the coefficient of variation for BP (BPCoV), cholesterol and triglyceride levels, and systolic BP from enrollment (baseline) to week 12.

### 2.5. Measurement of Blood Pressure

EHR-based blood pressure monitoring data are, by definition, obtained by a nurse in a clinic, whereas home monitoring data are provided by patients using a mobile device. Those measurements are likely to be somewhat discrepant [43]. In order to evaluate such discrepancies and determine the average degree of such variation between methods, the first (enrollment) visit, in which patients consented, included providing a home monitor (same monitor is used for all patients), training the patient in its use, and asking the patient to obtain a blood pressure reading (the second reading was used) from that monitor right after the same patient’s blood pressure was obtained by the nurse in the clinic using the clinic’s equipment. Only clinically obtained blood pressures were recorded in the EHR; the home monitoring data were considered research information (though they were disclosed to the provider, with patient’s agreement, in case of worrisome results as detailed in the study protocol). For the purpose of our analyses, we evaluated fluctuations and changes in systolic blood pressure (SBP, primary).

### 2.6. Statistical Analysis

The analyses were conducted using both summary statistics and hypothesis testing. Standard descriptive statistics were used to summarize the study data. Descriptive statistics such as the number of observations, the mean, the median, the 90% confidence interval, the standard deviation, the standard error, the minimum, and the maximum were considered for continuous variables. Other descriptive statistics such as counts, proportions, and/or percentages are presented to summarize discrete variables. Confidence intervals were estimated or hypothesis testing was performed for select endpoints, as described below. Exact binomial 90% confidence intervals for proportions are presented. Hypothesis testing was performed at the one-sided α = 0.05 significance level. No adjustments for multiplicity were planned. Demographic and baseline characteristics are summarized and used as determined by the study team (based on the distribution of those variables amongst the study population) in the sensitivity analyses. The analysis populations vary by specific aim/analysis. For example, the analysis of BP fluctuation changes pre–post intervention require at least 3 readings taken in the 6 months before and 3 months after implementation.

### 2.7. Analysis of Outcomes

Calculations were carried out for the proportions of (1) ‘at-risk’ patients identified via their EHR at baseline; (2) a subset of these ‘at-risk’ patients who were recruited to participate; and (3) those recruited ‘at-risk’ patients who successfully completed at least three remote intervention sessions over a 12-week period and whose associated 90% confidence intervals were reported. BP variability was evaluated through the BPSDV and the BPCoV. These metrics were based on measurements taken at weeks 2, 4, 8, and 12 and were compared using paired *t*-tests. Similarly, paired *t*-tests were used to assess changes in cholesterol and triglyceride levels from the initial visit to the final visit at week 12. We expected the enrolled patients to be relatively stable; it is possible that their blood pressure and other indicators would be within normal ranges at enrollment or during the intervention. Our main interest therefore was the reduction in fluctuation. Pre-intervention fluctuation was computed using the enrollment “variability” inclusion criterion, which is based on measurements of blood pressure for the full year pre-enrollment, or, in sensitivity analyses, 6-month pre-intervention values (to make the pre- and during intervention timeframes similar). If fewer than 3 readings were available within an analysis timeframe, the subject was excluded from the specific variability analysis, as the variability calculations required at least 3 values for the present trial.

### 2.8. Sample Size Consideration

This is a pilot study designed to assess feasibility and to provide an exploratory evaluation of the intervention’s effects on medical outcomes. With 20 patients, the exact binomial 90% CI for a proportion of 10% would be (1.8%, 28.3%), for 20%, it would be (7.1%, 40.1%), and for 50%, it would be (30.2%, 69.8%). These power estimates are provided to illustrate that the sample size is large enough to provide reliable estimates of the variance of continuous measures, which can be used in designing a larger study [44], allowing us to detect modest pre–post changes in continuous measures. It is acknowledged that this is not a definitive study but rather an exploration. The inclusion of twenty patients provides approximately 80% power to detect a change of 0.58 standard deviations at the one-sided 0.05 significance level.

## 3. Discussion

In line with its proof-of-concept focus, the CP&R intervention sets out to evaluate the feasibility and potential medical benefits of a remote intervention aimed at patients with high BP variability. It is, therefore, a study of secondary prevention (preventative efforts aimed at patients who are already known to be at risk). A large retrospective study [10] found that of 221,803 adults with at least two primary care visits over two years, 22.9% showed a high BP variability (pre-defined as SBP SD > 13 in that investigation). This rate increased to 33.2% among the subset of 85,455 hypertensive patients. Notably, high BP variability was twice as common in hypertensive individuals as in those without hypertension (33.2% vs. 16.5%, *p* < 0.001). Other investigations established that high BPV is associated with poor cardiovascular outcomes, even in the presence of otherwise “normal” mean blood pressure readings [5]. These findings underscore the need to include BP variability in risk assessments because simply looking at one measurement might lead to patients who are at risk being missed. Consistent with those findings, the CP&R intervention is the first prospective investigation, to our knowledge, to use a high level of variability (not “all comers”) as the entry criterion. As such, one of our most important outcomes would be to evaluate whether such patients can be identified, will give consent, and can be engaged in a behavioral intervention.

The effective control of BP plays a critical role in reducing the risk of serious complications like ischemic stroke and systemic embolism [7]. The analysis of pooled data from the CANVAS and CREDENCE trials suggests that in patients with type 2 diabetes, visit-to-visit variability in systolic BP is independently associated with an elevated risk of hospitalization for heart failure and all-cause mortality [6]. Similarly, research has shown that patients with chronic kidney disease who exhibit greater variability in systolic BP are at a heightened risk of major adverse cardiovascular events, which includes nonfatal myocardial infarction, unstable angina, revascularization, nonfatal stroke, heart failure, and cardiac death [8]. But BP variability is a complex phenomenon that has been hypothesized to be caused by multiple factors, including age, arterial stiffness, BP measurement errors, and seasonal changes [11]. Factors related to medication intake, such as improper dosage and poor adherence, are likely to play an important role as well. A post hoc analysis of the ALLHAT trial investigated the relationship between medication adherence and visit-to-visit BP variability [45]. This analysis focused on 2912 nonadherent and 16,878 adherent participants who had attended a minimum of five out of seven study visits between 6 and 28 months post-randomization. Participants were categorized as nonadherent if they self-reported taking less than 80% of their prescribed antihypertensive medication at one or more study visits. This study found a significant association between medication nonadherence and higher BP variability. Furthermore, changes in medication adherence directly correlated with changes in BP variability, highlighting the importance of improving adherence as a strategy for managing BP variability. In the CP&R trial, we posit that a very high level of variability is likely to be the result of less consistent medication-taking behavior over time. Trained interventionists engage with patients through telephone or internet chat to discuss medication-related concerns, review symptoms, and assist in implementing strategies to improve adherence. If successful, we expect to see a significant reduction in BPV during the study intervention period, as compared to before that.

Optimizing adherence to the standard of care has the potential to have a very significant impact on health. Although, in general, barriers to adherence might include medication costs [37], almost all medications used to control hypertension or hypercholesterolemia nowadays are generic. As a result, the standard of care for these conditions is largely covered by insurance carriers (whether commercial or others), and costs, therefore, tend to be a less important consideration as compared with patient behavior or general circumstances, including beliefs, psychological barriers, and the availability of reminders, monitoring, and support [3]. Indeed, a recent trial showed that optimizing the use of existing medications (rather than employing new ones), in the absence of any financial support, can have a profound effect on the success of treatment for cardiovascular illness in diverse populations worldwide [46].

Nevertheless, systematic attempts to improve patient engagement and adherence are very frequently ineffective at improving clinical outcomes, even though they might involve significant efforts [47,48]. One of the reasons for such frequent failure (and the reason such interventions are not generally incorporated into practice even if proven promising) is what we have termed “the streetlight effect”; adherence interventions often attempt to address the entire clinic rather than identify and target patients with erratic medication-taking behavior [49]. Such attempts (counter to a “precision medicine” approach) are likely to be inefficient (as they offer an intervention, such as education, to all comers). More importantly, without intentional efforts to specifically include nonadherent patients, intervention studies are likely to inadvertently not select them; the most nonadherent patients may attend clinic appointments less frequently and may therefore be less likely to be captured in recruitment efforts or to consent to participate in research when approached. They may also be less likely to engage in elaborate and time-consuming interventions.

Several innovations in our design address such shortcomings. First, our “precision medicine” approach only includes patients with demonstrable risk, as determined by an objective measure that can be obtained via manipulation of available EHR information obtained during routine care rather than conferring any burden or cost. Second, the intervention is delivered remotely and includes an innovative aspect that addresses medication avoidance, a construct largely overlooked in adherence research. This remote approach is particularly salient, as cardiology patients experience a substantial burden of disease and are at high risk for severe cases of infectious diseases like COVID-19 that can be contracted during a visit to a medical clinic [50]. Our approach is designed to reduce patient burden. But while burden is decreased by the remote nature of the intervention, the intervention itself is much more specific and intensive than, for example, automated reminders. It is conceivable that our intervention could be delivered by artificial intelligence (AI) in the future, but we do not believe that the care of at-risk patients, at this point in time, can be relegated to AI algorithms. Limitations to our design include the lack of a control group, the small sample size, and the limited follow-up time—all of which will have to be addressed in a larger, definitive trial that can follow the methodology of this exploratory trial. This exploratory trial will provide important information for a definitive, controlled trial, if warranted.

If proven feasible and potentially effective, a “tie-in” of the EHR algorithm to a remotely delivered intervention could represent a paradigm shift in behavioral risk assessment procedures as they are now practiced for the following reasons. First, the risk assessment does not require a clinician’s assessment. Second, the level of fluctuation can be continually assessed and monitored, and therefore the marker of risk can serve both as a way to “flag” patients at risk and as a way to monitor their progress. Third, as compared with a questionnaire or patient reports, the behavioral marker in this case is an objective assessment of behavior that is not subject to reporter bias. Because the intervention is remote, health systems could construct a central hub rather than requiring each individual system (or clinic) to develop the experience and expertise “in house”. We believe that if the results show improvement in patient outcomes, our approach would be not just economically feasible but in fact could be substantially more cost-effective than in-house behavioral health and patient coordination efforts.

## Figures and Tables

**Table 1 jcm-13-06274-t001:** Inclusion, exclusion, and withdrawal criteria.

Inclusion Criteria	
1	The patient is >18 years of age at enrollment.
2	The patient is monitored at the cardiology clinic, was diagnosed with hypertension and hypercholesterolemia more than 12 months prior to EHR screening, and has been prescribed at least one antihypertensive medication and at least one lipid-lowering agent over the 6 months prior to EHR screening.
3	The patient can be reached either by phone or via an internet-enabled device.
4	The patient speaks English or Spanish at a level that allows them to understand the study procedures and consent to the study.
5	The coefficient of variation (CoV) of at least three systolic blood pressure measurements present in the EHR over the 12 months prior to EHR screening is >10% [35].
**Exclusion Criteria**	
1	The patient is suffering from a psychiatric or developmental disorder that prevents him or her from understanding the protocol or engaging in the intervention (e.g., autistic disorder, psychosis).
2	The investigator determines that a remote intervention paradigm is not advisable because of specific patient or environmental characteristics (investigator discretion).
3	The patient is suffering from a medical disorder that makes control of blood pressure especially challenging or unlikely (e.g., end-stage renal disease, uncontrolled endocrine disorders) [38,39].
4	Unstable blood pressure or hyperlipidemia that may require change in therapy in the 3 months after enrollment [40].
5	Significant heart failure (NYHA > 2) or ejection fraction < 50% [41].
6	Recent thromboembolic events such as a myocardial infarction, stroke, acute coronary syndrome, or transient ischemic attack in the 6 months prior to enrollment
7	Any arrhythmia requiring medical or device therapy within 6 months prior to enrollment.
8	The patient is hospitalized or was hospitalized in the last 6 months prior to enrollment. Patients hospitalized after enrollment are not excluded.
**Withdrawal Criteria**	
1	The patient dies.
2	The patient becomes psychotic as defined in DSM-V or suffers from an event that makes him or her unable to participate in the intervention (e.g., loss of hearing, loss of cognitive ability).
3	The patient’s care is transferred to another center, and it is impossible to obtain the primary and secondary outcome data.
4	Patient decision.
5	Investigator decision.
6	DSMC or IRB determination related to the occurrence of an adverse event or for any other reason.

## Data Availability

Restrictions apply to the datasets due to ethical and privacy concerns.

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
