# Peer review of "Cardiovascular Precision Medicine and Remote Intervention Trial Rationale and Design"

_jcm, 2024, doi:10.3390/jcm13206274_

Round 1

Reviewer 1 Report

Comments and Suggestions for Authors

Dear Sirs,

I would like to thank you very much for sending me the manuscript entitled:

CP&R (Cardiovascular Precision Medicine & Remote Intervention) Trial Rationale and Design.”

At the outset, I would like to sincerely congratulate the authors on the work you have put into preparing the submitted manuscript.

The aim of this study is to search for a new risk marker by identifying patients through an algorithm linked to EHRs. The authors developed the methods used in an innovative ‘proof of concept’ study. A blood pressure variability index was calculated for clinic patients by reviewing the EHR. Consenting patients with excessive variability were offered a remote intervention to improve medication adherence. Outcomes included the ability to identify and engage identified patients and the effect of the intervention on variability, using a pre-post comparison design without a concurrent control. The authors concluded that the remote intervention reduces patient burden and promotes equitable and targeted resource allocation, using an objective criterion of behavioural risk.

In my opinion, the study will be of interest to readers

Yours sincerely,

Reviewer

Author Response

REVIEWER 1

Dear Sirs,

At the outset, I would like to sincerely congratulate the authors on the work you have put into preparing the submitted manuscript.

Thank you

In my opinion, the study will be of interest to readers

Thank you

Reviewer 2 Report

Comments and Suggestions for Authors

1. This is a well written paper on Blood pressure variability and description of CP & R trial design. It is important to note ESRD patients, post MI , stroke patients  in first 6 months, patients with mental illness are excluded. This high risk group has more chances of BP variability and this can have adverse impact on them. I was curious to know what was the reason they were excluded?

Comments on the Quality of English Language

1. Minor spelling errors in Exclusion criteria--enrollment, Arrhythmia.

Author Response

REVIEWER 2

This is a well written paper on Blood pressure variability and description of CP & R trial design.

Thank you

It is important to note ESRD patients, post MI , stroke patients  in first 6 months, patients with mental illness are excluded. This high risk group has more chances of BP variability and this can have adverse impact on them. I was curious to know what was the reason they were excluded?

Thank you.  We now added that we specifically wanted to investigate a group of patients that do not have such indicators and therefore may not otherwise be considered “high risk” and in need for more intensive surveillance.  Our paradigm is “case identification” through the variability metric and not via other features that might be recognized by practitioners anyway.

Comments on the Quality of English Language

  1. Minor spelling errors in Exclusion criteria--enrollment, Arrhythmia.

Thank you, fixed.

Reviewer 3 Report

Comments and Suggestions for Authors

I have read with great interest the manuscript by Reynolds et al, regarding the rationale and study design of CP&R (Cardiovascular Precision Medicine & Remote Intervention). Blood pressure variability, both short- and long-term seems to associate both with hypertension-mediated organ damage (HMOD) as well as with cardiovascular prognosis and “hard” endpoints. However, little is known about the role of BP variability in distinguishing patients being at increased risk through an electronic surveillance by modern telemonitoring systems.

In Introduction you can also add this reference (https://pubmed.ncbi.nlm.nih.gov/35322181/) that between others provides novel knowledge regarding BP and BP variability assessment by wearables and telemonitoring technology.

In Introduction, Lines 34-38 need reference (Who data, Guidelines ESC, AHA). The rest of introduction is an excellent form of how introductions should be written when addressing the rationale of a trial. The authors provide evidence, as well as the literature gap in Lines 68-70 that this trial should try to evaluate.

In Lines 92-93 can you specifically mention the “termination of the intervention”? – how long is the follow-up.

Is hospitalization for cardiovascular reason or hospitalization for non-cardiovascular cause issue regarding exclusion from the study? Or terminating a patients’ enrollment?

This is a proof-of-concept trial, therefore I did not anticipate power analysis. Although in Lines  185-187 you mention that with 20 patients, the exact binomial 90% CI for a proportion of 10% would be (1.8%, 28.3%), for 20% would be (7.1%, 186 40.1%), and for 50% would be (30.2%, 69.8%), will you enroll only 20 ? Can you mention in your design if you anticipate a more optimistic number, regarding enrolled patients?

Finally, I am very grateful that you mentioned in the end of your discussion possible limitations of your study? Can you add a paragraph afterwards, regarding possible changes in future trial design you could make, if your proof-of-concept study is successful?

Again, I am very thrilled that I read such an interesting trial design, as I am also keen on AI, wearables, telemonitoring and EHR in cardiovascular medicine

Author Response

REVIEWER 3

I have read with great interest the manuscript by Reynolds et al, regarding the rationale and study design of CP&R (Cardiovascular Precision Medicine & Remote Intervention). Blood pressure variability, both short- and long-term seems to associate both with hypertension-mediated organ damage (HMOD) as well as with cardiovascular prognosis and “hard” endpoints.

Thank you

In Introduction you can also add this reference (https://pubmed.ncbi.nlm.nih.gov/35322181/) that between others provides novel knowledge regarding BP and BP variability assessment by wearables and telemonitoring technology.

This reference is indeed helpful, as it mentions a novel way of calculating variability more readily.  We have included it in the introduction as suggested.  Thank you.

In Introduction, Lines 34-38 need reference (Who data, Guidelines ESC, AHA).

WHO reference added, thank you

The rest of introduction is an excellent form of how introductions should be written when addressing the rationale of a trial. The authors provide evidence, as well as the literature gap in Lines 68-70 that this trial should try to evaluate.

Thank you

In Lines 92-93 can you specifically mention the “termination of the intervention”? – how long is the follow-up.

We now specify that “termination of the intervention” means the time of the exit session, which is at 12 week post-enrollment’s beginning session, with a 2 month “window”.  We also added a clarification about the timeframe of different analyses using pre-intervention data.

Is hospitalization for cardiovascular reason or hospitalization for non-cardiovascular cause issue regarding exclusion from the study? Or terminating a patients’ enrollment?

Patients cannot be enrolled while in-patient or if they were hospitalized within the 6 months prior to enrollment, but hospitalization during the study is not reason for withdrawal.  The reason for exclusion of in-patients at time of enrollment or within the 6 months prior is that we wanted to recruit patients who are supposedly “stable” regarding their apparent clinical course, but “unstable” per the fluctuation marker.  Those patients would not appear to be in need of enhanced monitoring and frequent contact, and yet, they probably are in need of enhanced care (according to the fluctuation marker).  We thank the reviewer for bringing this up.  This point is now explained in the text.

This is a proof-of-concept trial, therefore I did not anticipate power analysis. Although in Lines  185-187 you mention that with 20 patients, the exact binomial 90% CI for a proportion of 10% would be (1.8%, 28.3%), for 20% would be (7.1%, 186 40.1%), and for 50% would be (30.2%, 69.8%), will you enroll only 20 ? Can you mention in your design if you anticipate a more optimistic number, regarding enrolled patients?

We agree that a power analysis is not required for such a trial, the point of explaining our power consideration is just to show that the study will be able to provide sufficient information about variance in the metrics that we have chosen, in order to inform a more robust power analysis for a later (controlled) trial.  This point is now incorporated into the text.  We believe that our sample size is well-considered and we do not plan to recruit a larger number of patients than stated.  The number of analyzable subjects that we aim at is expected to give us a reasonable idea about the intervention’s potential, but of course, this is not going to be a definitive study.

Finally, I am very grateful that you mentioned in the end of your discussion possible limitations of your study? Can you add a paragraph afterwards, regarding possible changes in future trial design you could make, if your proof-of-concept study is successful?

Certainly, thank you – we added that a controlled trial is going to have to be the next step if this study looks promising, and that we will use the feasibility information to determine the number of sites involved and length of the intervention.

Again, I am very thrilled that I read such an interesting trial design, as I am also keen on AI, wearables, telemonitoring and EHR in cardiovascular medicine

Thank you

Reviewer 4 Report

Comments and Suggestions for Authors

The submitted paper is not an original article but instead a study protocol. The study protocol is in the field of preventive medicine and aims at exploring BP variability between visits and at home  with a pre-post comparison after a telemedicine intervention. Although the results of the study will be of clear interest, I do not believe this study protocol is in the aims of the Journals. Furthermore, a schematic representation of the study design is missing and several applications have been developed to monitoring patient compliance to medications. A remote contact every one or two week is not feasible in most health systems. 

Author Response

REVIEWER 4

The submitted paper is not an original article but instead a study protocol. The study protocol is in the field of preventive medicine and aims at exploring BP variability between visits and at home  with a pre-post comparison after a telemedicine intervention.

We agree with this characterization.  The “prevention” here is secondary (not primary) prevention.  We incorporated this insight into the text, thank you.  We note that since we are actively assessing cardiovascular risk and it is an inclusion criterion, the relevant discipline is cardiovascular medicine.

Although the results of the study will be of clear interest, I do not believe this study protocol is in the aims of the Journals.

We thank the reviewer for their interest in the results, and we believe that our study aims and design are sufficiently unique so as to be of interest to the readers, as further exemplified in our responses to the reviewer’s next comment (below).

Furthermore, a schematic representation of the study design is missing

This was added.

and several applications have been developed to monitoring patient compliance to medications.

Although applications are being developed, to our knowledge none are satisfactory to date and most still rely on patient report in one or other forms – ours is not an “application” but a novel way to analyze existing EHR data which is not at all the same as electronic apps.  We pre-select patients at high risk rather than provide a generic monitoring scheme, and the intervention actually involves speaking with the few identified patients who are at risk, routinely and repeatedly.  So this monitoring / intervention scheme relies on EHR data to inform a very targeted intervention that presumably is more likely to improve outcomes for patients who are truly at risk and hard to engage.  Those features are – in our view – innovative and therefore of interest to readers. 

 A remote contact every one or two week is not feasible in most health systems. 

We appreciate the comment but beg to differ.  The intervention is only applicable for a small fraction of patients who are “flagged” by EHR review.  Those patients are at significant risk. 

The fact that the screening is done using data readily available in the EHR means that it would be quite easy to implement this screening plan, in any health system that uses EHR.  And there is no increase in patient burden to obtain this information.  The fact that only a small number of patients are identified for intervention inclusion means that the intervention effort is specific, targeted, and efficient.  Because the intervention is remote, a centralized approach could be implemented, in which health systems could contract a central hub rather than requiring each individual system (or clinic) to develop the experience and expertise “in house”.  We believe that, if the results show improvement in patient outcomes, our approach would be not just economically feasible but in fact could be substantially more cost-effective than in-house behavioral health and patient coordination efforts.  As generally used nowadays, such efforts do not involve a significant targeting mechanism and require the presence and training of in-house specialists (such as social workers, psychologists, etc.).  We have made these points more explicit in the revised text so the innovative aspect of the intervention is clearer to the reader.

We appreciate the reviewers’ comments and believe that the resulting revised manuscript states more clearly the true innovation and promise in this design – thank you.

Round 2

Reviewer 4 Report

Comments and Suggestions for Authors

I appreciate Authors response to this Reviewer comments.